# On-farm evaluation of growth and reproductive performances of Washera and Gumuz sheep in northwestern Ethiopia: Basics for setting up breeding objectives/goals

Sisay Asmare[1]☯*, Kefyalew Alemayehu[2‡], Solomon Abegaz[3‡], Aynalem Haile[4‡]

1 Debre Markos University, Burie Campus, Debre Markos, Ethiopia, 2 Department of Animal Science and Technology, College of Agriculture and Environmental Sciences, Bahir Dar University, Bahir Dar, Ethiopia, 3 Ethiopian Institute of Agricultural Research, Addis Ababa, Ethiopia, 4 International Center for Agricultural Research in the Dry Areas (ICARDA), Addis Ababa, Ethiopia

☯ These authors contributed equally to this work.
‡ These authors also contributed equally to this work.
* sisayasmbel@yahoo.com

## Abstract

Growth and reproductive performance traits are traits of economic importance for sheep selection and productivity improvement interventions. This study aimed at comparative evaluation of growth and reproductive performance traits of sheep in the highland and lowland agro-ecologies of northwestern Ethiopia. Data on growth performance traits were collected from 144 Washera (78 males and 66 females) lambs and 72 Gumuz (37 males and 35 females) lambs. Data on reproductive performance traits were collected from 260 Washera (130 rams and 130 ewes) sheep and 150 Gumuz (75 rams and 75 ewes) sheep. General linear model univariate procedure was employed to analyze the collected data. Breed, the interaction effect between breed and season of birth as well as the interaction effect between breed and type of birth all exerted very high significant effect (P<0.001) on live weight at all age groups. Breed type affected pre-weaning average daily weight gain significantly (P<0.01). Pre-weaning average daily weight gain of Washera (70 g/day) was found much better performance than 60 g/day of Gumuz. Breed type exerts significant (P<0.05) effect on age at first lambing, lambing interval, annual reproductive rate and number of lambs born per ewe life time. Average age at first lambing and lambing interval of Washera sheep were 11.69 months and 9.27 months, respectively. The corresponding values for Gumuz sheep were 12.51 months and 10.43 months, respectively. Production and reproduction performance values of traits varied across the two breeds and sexes as well. These values can be used to set up breeding objectives or goals for selective breeding of sheep giving special emphasis to growth traits believed to have medium heritability values.

**Data Availability Statement:** All relevant data are within the manuscript and its Supporting Information files.

**Funding:** Sisay Asmare received fund from Biotechnology Research Institute of Bahir Dar University by an award number of 1/8048/14-10. The website for College of Agriculture and Environmental Sciences of Bahir Dar University is https://www.bdu.edu.et/bri/. The funders do not take part or had no role in study conceptualization and design, data collection and analysis, decision to publish, or preparation of the manuscript.

**Competing interests:** The authors have declared that no competing interests exist.

# Introduction

Since domestication, sheep have become essential parts of farms across the world [1] and comprises 16 percent of the total tropical livestock unit (TLU) in sub-Saharan Africa [2]. In part it is because of their ability to adapt to local environments [1]. Small ruminants represent only 7% of the average total capital invested in livestock in the mixed crop-livestock production system, but they account on average for 40% of the cash income and 19% of the total value of subsistence food derived from all livestock production [3, 4]. Sheep and goats contribute a quarter of the domestic meat consumption; about half of the domestic wool requirements, about 40% of fresh skins and 92% of the value of semi-processed skin and hide export trade. The increased domestic and international demand for Ethiopian sheep and goats has established them as important sources of Inland Revenue as well as foreign currency [3]. Therefore, integrated attempt with emphasis on prescribing and genetic enhancement is crucial for improving animal output by improving growth and fertility traits [5]. In Ethiopia, there are 32.85 million sheep, more than 99% of which are indigenous [6]. However, the production of local sheep under traditional production system is low with high mortality of lambs [7]. Because of this the increasing need for food of animal origin has largely been met with increasing number of sheep while productivity per sheep has remained low [8]. Live weight and growth rate are economically critical features, requiring particular attention in any breeding program intended to improve overall productivity since lambs are mainly raised for mutton [7]. Besides, good reproductive performance is a prerequisite for any successful genetic improvement and it determines production efficiency which depends on various factors including age at first lambing, litter size, lambing interval and the life time productivity of the ewe [9–11]. Indigenous sheep breeds are compatible genotypes for the low input traditional production system since indigenous sheep breeds have special adaptive features such as tolerance to a wide range of diseases, water scarcity tolerance and ability to better utilize the limited and poor quality feed resources [12]. Conservation and utilization of these local breeds of sheep is part of animal husbandry and should, ideally, be based on complete information on distribution, structures, and trends, productive and adaptive performances of populations of the existing breeds [13].

Washera and Gumuz indigenous sheep were produced under highland cereal-livestock and lowland crop-livestock sheep production systems, respectively [14]. Morphologically, Washera is a short fat tail, large body size, short-haired, predominantly brown, both males and females are polled reared by Amhara and Agew communities. Washera is a good meat producer under good environment [15]. Gumuz sheep is long thin tail, somewhat dwarf, convex face profile, long pendulous ear, commonly plain brown or with patch coat color, polled and reared by Gumuz and Amhara communities [14]. Gumuz sheep is adapted to heat and have unique genetic make-up [15]. On-farm monitoring involves monitoring the productive and reproductive performance of a breed on selected representative village flocks or herds [16]. An earlier study was conducted on growth performance of Washera sheep in Quarit and Yelmandensa districts of Amhara region, Ethiopia [17]. Nevertheless, most Ethiopian sheep breed types are characterized by high within-population variability, have undergone little selection for improved meat production and true breed potentials are not known [18, 19].

In Ethiopia, researches regarding on-farm performance of indigenous sheep and their crosses with exotic breeds are not well grounded due to the reason that many researchers has tended to focus on-station performance evaluation rather than on-farm condition [20]. Yet, on-farm performance evaluation gives a more representative performance level of the breed since it is undertaken under the natural production environment of the breed [16]. Thus, the objective of this study was to evaluate growth and reproductive performance of Washera and Gumuz indigenous sheep and generate information for further establishment of selective

breeding and productivity improvement programs under low input sheep production systems of highlands and lowlands of northwestern Ethiopia.

## Materials and methods

### Ethics statement

Earlier to the study, data collection formats and procedures were reviewed and agreed by the research ethics review committee of Debre Markos University, Ethiopia (Ref. number DMU 10/2018 E.C). In addition, target group smallholder farmers provided their verbal informed consent to participate in this study. Besides, the data were analyzed namelessly; ethnicity and religious issues were not asked and recorded during data collection.

### Geographic coordinates of the study areas

This study was conducted in two districts, Burie and Mandura considered as potential areas for Washera and Gumuz sheep breed types in North Western Ethiopia. According to altitudinal location, rainfall and crop yields, the agro-ecological zone of Burie is highland and the agro-ecological zone of Mandura is lowland [21].

### Sample size and sampling methods

After preliminary survey, two districts Burie and Mandura were selected by purposive sampling method [22] based on sheep production potential. Similarly, two rural peasant associations (Wanegedam and Daez Baguna) were selected from their respective districts. The key informants including the development agents and peasant association administrators were given training to enumerate smallholders engaged in sheep production keeping three sheep or more and how to fill questionnaires.

In the mean time, from a total of 1103 households of Burie district engaged in sheep production, 260 households owning ram and ewe giving birth at commencement of data collection were selected by random sampling method. A total of 260 Washera (130 rams and 130 ewes) sheep were considered for the monitoring of reproductive performance. A total of 144 Washera (78 males and 66 females) lambs were considered for the monitoring of growth performance.

Similarly, from a total of 622 households of Mandura district engaged in sheep production, 150 households owning ewe giving birth at commencement of data collection were selected by random sampling method [22]. A total of 150 Gumuz (75 rams and 75 ewes) sheep were considered for the monitoring of reproductive performance. A total of 72 Gumuz (37 males and 35 females) lambs were considered for monitoring growth performance.

### Data collection and animal management

The necessary data on growth and reproductive parameters were collected from sheep flocks managed under mixed—crop livestock farming system of Burie and Mandura districts. Data collection was carried out for two years from September, 2017 to October, 2018 G.C. Smallholders owning rams, ewes and lambs were asked for voluntarily for sparing their sheep and time while recording monitoring data on growth and reproductive performance traits. Live weight measurements were taken five times at the age of day 1, 90 days, 180 days, 270 days and 365 days. Live weight measurements were taken in the mornings, before the animals left the shelter to graze to avoid undesirable variations because of changes in rumen volumes [23, 24]. Lamb weight measurements were taken taking ethical and animal welfare into account. Suspended weighing scale, rope, tripod stick and soil fertilizer bags were purchased from Bahir

Dar city general market and were used for measuring live weight of lambs. Concurrently, date of birth, birth weight, type of birth, sex of lamb, season of birth, lambing interval, age at first lambing, age at first mating, number of lambs born during life time of the ewe, reproductive age of the ewe, number of offspring produced, number of offspring weaned and ewe parity were recorded starting from 24 hours of the new born by the recruited enumerators in each respective study site. During the course of data collection, dams and lambs were managed to graze in communal grazing lands and crop aftermaths for an average of eight hour and supplemented with crop residues and *atella* (bi-product of local brewery). Different formulas were used to calculate some parameters.

Weight records at various ages were adjusted for actual age using different formulas used by [25, 26].

And average daily weight gain was calculated by the formulas:

$$\text{Average daily weight gain from birth to weaning (g)} = (\text{AWWT(kg)} - \text{BWT (kg)})/\,90*1000 \tag{1}$$

$$\text{Average daily weight gain from weaning to 6 month age (g)} = (\text{A6MWT(kg)} - \text{AWWT(kg)})/\,90*1000 \tag{2}$$

$$\text{Average weight gain from 6 month age to 365 days yearling (g)} = (\text{AYWT9(kg)} - \text{AWWT(kg)})/\,275*1000 \tag{3}$$

Where:
BWT = Birth weight
AWWT = Adjusted weaning weight at 90 days
A6MWT = Adjusted 6 month weight at 180 days
AYWT = Adjusted yearling weight at 365 days
Annual reproductive rate (ARR) of Washera and Gumuz ewes was calculated by formulae adopted from [27]:

$$\text{Annual reproductive rate (ARR)} = 365 * \text{average litter size/average days of lambing interval} \tag{4}$$

Lamb survival rate was calculated by the formulae adopted from [28]:

$$\text{Lamb survival rate (\%)} = (\text{Number of offspring weaned/ number of offspring produced}) *100 \tag{5}$$

## Statistical analysis

The data collected on growth and reproductive performance traits were arranged in an excel spread sheet. Ultimately, the data were analyzed using General Linear Model(GLM) univariate procedure of Statistical Package for Social Science for window version 20.0, SPSS (2011) [29] fitting growth rate traits (birth weight, adjusted weaning weight, adjusted 6 month weight, adjusted yearling weight, pre weaning and post weaning average daily weight gain) and reproductive performance traits such as age at first lambing, lambing interval, annual reproductive rate, number of lambs born per ewe life time as dependent variable and non-genetic factors such as breed, sex of lamb, parity of the dam, season of birth and type of birth as fixed factor (s). Non significant effects of independent variables (non-genetic factors) were not reported.

Data analysis of growth and reproductive performance traits were carried out using two different statistical models:

The statistical model for analyzing growth performance was written as follows:

$$Y_{ijklmno} = \mu + B_i + S_j + P_k + S_l + T_m + (B_i x S_l)_n + (B_i x T_m)_o + e_{ijklmno} \qquad \text{(model 1)}$$

Where:

$Y_{ijklmno}$ = the body weight and average daily weight gain of the $n$th lamb of $n^{th}$ growing lamb

$\mu$ = population mean

$B_i$ = effect of $i^{th}$ breed (Washera, Gumez)

$S_j$ = effect of $j^{th}$ sex of lamb (male or female)

$P_k$ = effect of $k^{th}$ parity of the dam (P = first, second, third and fourth)

$S_l$ = effect of lth season of birth (B(*belg*) = short rainy season (March–May), S = cold dry season (September-November), DS = Dry Season (December-February), RS = Rainy Season (June-August)

$T_m$ = effect of $m^{th}$ type of birth (single, twin)

$B_i x S_l$ = interaction effect of $i^{th}$ breed and $l^{th}$ season of birth

$B_i x T_m$ = interaction effect of $i^{th}$ breed and $m^{th}$ type of birth

$E_{ijklmno}$ = error/residual effect

The statistical model for analyzing reproductive performance was written as follows:

$$Y_i = \mu + B_i + e_i \qquad \text{(model 2)}$$

Where:

$Y_i$ = reproductive performance traits (age at first mating of males, age at first mating of females, age at first lambing of females, lambing interval of females, annual reproductive rate of females, number of lambs born per ewe life time of females etc)

U = population mean

$B_i$ = the effect of $i^{th}$ breed (Washera, Gumuz)

$E_i$ = Error/residual effect

## Results and discussion

### Growth performance of Washera and Gumuz sheep

Live weight of Washera and Gumuz sheep populations were analyzed for different age groups. Table 1 depicts live weight in kg at different age groups and the effect of factors associated with. Breed type affects pre-weaning average daily weight gain of lambs at very high significant level (p<0.001). Pre-weaning average daily weight gain were 70 g/day and 60 g/day for Washera and Gumuz lambs, respectively. There is no significant difference in pre-weaning average body weight gain of lambs across the two sexes. The overall post weaning average daily weight gain of lambs from 6 to 12 months of age was 50 g/day. Parity and season of birth all have very high significant (p<0.001) effect on post weaning average daily weight gain of Washera and Gumuz lambs.

### Reproductive performance of Washera and Gumuz sheep

Measures of reproduction commonly used in sheep and goats include age at puberty, age at first lambing/kidding, post-partum interval, and parturition interval and fertility indices [10]. Table 2 depicts least squares mean values for some reproductive traits of Washera and Gumuz sheep and the effect of breed/ecotype.

**Table 1. Least squares means of birth, weaning, six month, yearling weight and tests of between subject effects.**

| Factors | N | BWT | WWT | 6MW | YW |
|---|---|---|---|---|---|
| | | Mean±SD | Mean± SD | Mean± SD | Mean± SD |
| **Breed** | | *** | *** | *** | *** |
| Overall | 216 | 1.98±0.60 | 8.22±0.11 | 13.35±1.16 | 21.33±2.00 |
| Washera | 144 | 2.16±0.54 | 8.59±0.79 | 13.94±1.2 | 21.8±1.91 |
| Gumuz | 72 | 1.62±0.55 | 7.45±1.26 | 12.16±1.67 | 20.37±1.85 |
| Sex of lamb | | *** | ns | * | * |
| Overall | 216 | 1.98±0.60 | 8.22±1.11 | 13.35±1.61 | 21.33±2.00 |
| Male | 115 | 2.19±0.52 | 8.41±1.09 | 13.87±1.47 | 22.08±1.78 |
| Female | 101 | 1.73±0.59 | 7.99±1.09 | 12.75±1.56 | 20.47±1.90 |
| Parity | | ns | ns | ns | *** |
| Overall | 216 | 1.98±0.60 | 8.22±1.11 | 13.35±1.61 | 21.33±2.00 |
| 1 | 74 | 1.78±0.66 | 7.75±1.14 | 12.34±1.46 | 19.67±1.12 |
| 2 | 67 | 2.15±0.54 | 8.63±0.79 | 14.24±1.23 | 22.9±1.33 |
| 3 | 40 | 2.08±0.53 | 8.49±1.13 | 14.09±1.24 | 22.98±1.35 |
| 4 | 35 | 1.94±0.53 | 8.08±1.16 | 12.9±1.56 | 19.9±1.03 |
| Season of birth | | ns | * | ** | *** |
| Overall | 216 | 1.98±0.60 | 8.22±1.11 | 13.35±1.61 | 21.33±2.00 |
| Short rainy season(March–May) | 54 | 2.12±0.53 | 8.88±0.63 | 14.07±0.86 | 21.68±0.93 |
| Rainy Season(June-August) | 74 | 2.08±0.57 | 8.42±1.10 | 14.23±1.37 | 23.36±1.13 |
| Cold dry season (September-November) | 32 | 1.85±0.57 | 7.75±1.14 | 12.6±1.62 | 20.44±0.79 |
| Dry Season (December-February) | 65 | 1.83±0.65 | 7.77±1.08 | 12.26±1.43 | 19.3±1.34 |
| Type of birth | | *** | * | ** | ns |
| Overall | 216 | 1.98±0.60 | 8.21±1.11 | 13.35±1.61 | 21.74±2.00 |
| Single | 92 | 2.48±0.60 | 8.57±1.15 | 13.95±1.49 | 21.74±2.00 |
| Twin | 124 | 1.6±0.60 | 7.95±1.08 | 12.89±1.70 | 21.02±2.00 |
| $B_i x S_l$ | 216 | *** | *** | *** | *** |
| $B_i x T_m$ | 216 | *** | ** | *** | *** |

SD = standard deviation, BWT = birth weight, WWT = weaning weight, 6MW = six month weight, YW = yearling weight, Bi = breed, SL = season of birth, Tm = type of birth, values, $B_i x S_l$ = the interaction effect between breed and season of birth, $B_i x T_m$ = the interaction effect between breed and type of birth

* = significant at p<0.05

** = significant at p<0.01

*** = very high significant at p<0.001, ns = non significant

**Table 2. Least squares means of reproductive traits and tests of between subject effects.**

| Factors | N | AFMM | AFMF | AFLE | ALIE | TB | ARAE | LBELT |
|---|---|---|---|---|---|---|---|---|
| | | Mean±SD | Mean± SD | Mean±SD | Mean±SD | Mean± SD | Mean± SD | Mean± SD |
| **Breed** | | *** | *** | ** | ** | ** | ns | *** |
| Overall | 410 | 11.0±0.2 | 6.9±1.5 | 11.9±1.9 | 9.6±2.9 | 2.1±1.3 | 6.6±1.7 | 11.4±4.1 |
| Washera | 260 | 12.2±0.2 | 6.5±1.3 | 11.6±1.9 | 9.2±3.5 | 2.4±1.5 | 6.6±1.8 | 12.2±4.5 |
| Gumuz | 150 | 9.8±0.5 | 7.6±1.6 | 12.5±1.9 | 10.4±1.2 | 1.7±0.6 | 6.7±1.4 | 9.8±2.7 |

SD = Standard deviation, AFMM = age at first mating of males, AFMF = age at first mating of females, AFLE = age at first lambing of ewes, ALIE = average lambing interval of ewes, TB = type of birth, ARAE = average reproductive age of ewes, LBELT = lambs born per ewe life time

* = significant at p<0.05

** = significant at p<0.01

*** = very high significant at p<0.001, ns = non significant

The impact of reproduction on sheep and goat productivity is best estimated by the annual reproductive rate [9]. Studies carried out over periods of several years have shown that the annual reproductive rate (number of young produced per breeding female per year) of African breeds varies from 1.5 to over two lambs [27]. This rate of reproduction results in part from the uncontrolled access of rams to ewes on a permanent basis and in part from the litter size, in that, annual reproductive rate (ARR) is a function of litter size and parturition interval (ARR = Liter size*365/ parturition interval. Annual reproductive rate was 2.12 and 1.71 lambs/ ewe/year for Washera and Gumuz sheep, respectively. Annual reproductive rate of Washera (2.12 lambs/ewe/year) and Gumuz (1.71 lambs/ewe/year) was within the range of [27]. Lamb survival rate were 70.7% and 70.1% for Washera and Gumuz sheep, respectively.

## Growth performance and the effect of factors

Mean±SD values of birth weight of Washera lamb was close to 2.2±0.04 kg of Wollo highland lambs [30] and 2.4±0.2 kg of Bonga sheep [31] under village management condition. Yearling weight values of Washera and Gumuz sheep were close to 23.7 ± 0.04 kg of Horro sheep [32].

Growth and reproductive performance of indigenous sheep could be improved through improvement of genetics and environment [5]. However, growth and reproductive performance of indigenous sheep was influenced by most of the non-genetic factors [33, 34] and it is thus appropriate to consider the factors affecting their outcome. The current study revealed similar result. For instance, breed, the interaction effect between breed and season of birth and the interaction effect between breed and type of birth all exerted significant effect (P<0.01) on live weight at all age groups. Sex affected live weight at all age groups significantly (P<0.05) except on weaning weight. Type of birth affects birth weight significantly (P<0.05). Larger birth weight was recorded for single born lambs than their twin contemporaries. This is due to the fact that single born lambs have got a better chance of getting good nourishment at the fetal stage. The effects of breed, sex and type of birth on birth weight were in agreement with the view of [7] for Horro and Menz sheep. Season of birth affects weaning weight significantly (P<0.05). Lambs born during rainy season (June-August) and short rainy season (March-May) of the year were found to be heavier than lambs born during cold dry season (September-November) and dry season (December-February). This could be due to feed availability during these seasons for the dam to meet its production requirement for lactation beyond maintenance requirement to suckle the lamb until weaning and to the lamb side to graze on green pasture from weaning onwards. Parity and season of birth were non-genetic factors exerting significant (P<0.001) effect on yearling weight. Larger yearling weight observed during 2nd and 3rd parities. Season of birth affects yearling weight significantly (P<0.001). Lambs born during rainy season (June-August) and short rainy season (March–May) of the year were found to be heavier than lambs born during cold dry season (September-November) and dry season (December-February). This could be due to feed availability during these seasons for the lamb to graze on green pasture. Breed type affected pre-weaning average daily weight gain significantly (P<0.01). The variations in pre-weaning average daily weight gain values of Washera and Gumuz sheep were in line with [19] report suggesting that wide variability exists among Ethiopian small ruminant breeds with respect to potential growth rates and mature weight. Type of birth affected pre-weaning average daily weight gain significantly (P<0.01). A similar view of this result was held by [7] for Horro and Menz sheep. The post-weaning growth rate of Washera sheep was comparable and even better than some other indigenous breeds indicating its potential for meat (mutton) production for the local and export markets [35]. For instance, post-weaning growth rate of Washera sheep found by this study was 60 g/day, which was close to 63.4±4.0 of Bonga sheep [32].

## Reproductive performance and the effect of factors

The overall mean±SD value for average age of first mating for females was 6.93±1.54 months (207.9 days). These ewes were early maturing than Djallonké gimmers of Ghana mated when they are 8 months (240 days) old [36]. Litter sizes of Washera (1.62 lambs/birth) and Gumuz (1.43±0.49 lambs/birth) sheep were higher than 1.34 ± 0.01 lambs/birth of Horro sheep [35] in South Western Ethiopia. Litter size values of Washera and Gumuz sheep indicated that these sheep populations were prolific and comparable to other Ethiopian and most African breeds. The overall mean±SD value for lambing interval of this study was 9.69±2.96 months (290.7 days). This lambing interval was longer than 262 ±53.4 days of sheep in South Western part of Ethiopia [33]. However, improving the management practice will make shorter parturition intervals (below 9.69±2.96 months) enabling approximately three lamb crops to be obtained in two years. Annual reproductive rate of Washera ewe (2.12 lambs/ewe/year) was found to be greater than 1.88 ± 0.44 of sheep in South Western part of Ethiopia [33]. Annual reproductive rate (ARR) of Gumuz ewe (1.71lambs/ewe/year) looks closer to 1.88 ± 0.44 of sheep in South Western part of Ethiopia [33]. Annual reproductive rate of Washera and Gumuz sheep managed under smallholder mixed farming system of North Western Ethiopia found to be better performance compared to 1.20 and 1.05 [27] for pastoral and agro-pastoral areas of Ethiopia, respectively. The overall mean±SD value for number of lambs born per ewe life time (LBELT) of Washera and Gumuz sheep was 11.4±4.13 lambs per ewe life time. Lambs born per ewe life time of Washera and Gumuz sheep were less than 12.06 lambs per ewe life time of sheep in pastoral sheep production systems of Ethiopia [37]. Lamb survival rate were 70.7% and 70.1% for Washera and Gumuz sheep, respectively. This comparable value of lamb survival rate of Washera and Gumuz sheep was attributed to similar village management condition of sheep and similar mixed crop-livestock sheep production system to which these indigenous breeds are kept. Lamb survival rate of Washera and Gumuz were higher than 68% of Horro sheep but less than 83% of Menz sheep [38]. Thus, the tendency of pre weaning lamb loss will be higher for Washera, Gumuz and Horro than for Menz sheep. This variation in lamb survival rate between Washera, Gumuz and Menz could be attributed to sheep management system within which these indigenous breeds were kept (on-farm versus on-station). Moreover, the variation could be explained by varied sheep production systems (mixed crop-livestock of Washera and Gumuz versus sheep barley system of Menz sheep).

Tests between subject effects indicated that breed type exerted very high significant (P<0.001) effect on age at first mating of males, age at first mating of females and lambs born per ewe life time. The influence of breed type on age at first lambing, lambing interval, and type of birth was significant at P<0.01. However, the effect of breed type on average reproductive age of ewes is not significant. Age at first lambing, lambing interval, annual reproductive rate and number of lambs born per ewe life time all vary across the two breed types. The effect of breed type on most reproductive traits of Washera and Gumuz was in line with the earlier report [31] for Bonga sheep in Southern Ethiopia. The opposite view is expressed by [33, 34] report suggesting that district has no influence on age at first lambing, lambing interval, annual reproductive rate and number of lambs born per life time of ewe traditionally managed in the South Western part of Ethiopia and Begayt sheep of northwestern Zone of Tigray Region, in Humera district, Northern part of Ethiopia.

## Conclusion

This study was aimed at evaluating growth and reproductive performance traits of indigenous sheep breeds under on-farm sheep management conditions of northwestern Ethiopia. Production and reproduction performance values for traits of Washera and Gumuz sheep were

comparable or greater than to other Ethiopian and African breeds/ecotypes. The variations between minimum and maximum values of growth and reproductive performance traits indicated the presence of within breed variation. The observed variations in values of growth and reproductive performance were essential steps for setting up breeding objectives or goals of sheep breeding program for selection within and among indigenous sheep populations in the highlands and lowlands of northwestern Ethiopia by giving special consideration to growth rate traits believed to have medium heritability values. In the mean time, these values could be used as a valuable input for selective breeding and productivity improvement.

## Supporting information

**S1 Data. Data of growth performance.**
(XLS)

**S2 Data. Data of reproductive performance.**
(XLS)

## Acknowledgments

The authors sincerely thank smallholder farmers of the study areas for participating in restraining and handling of their sheep while measuring morphometric traits. We also wish to thank development agents of respective district agricultural offices for their role in data collection.

## Author Contributions

**Conceptualization:** Sisay Asmare.

**Data curation:** Sisay Asmare.

**Formal analysis:** Sisay Asmare.

**Investigation:** Sisay Asmare.

**Methodology:** Sisay Asmare.

**Project administration:** Sisay Asmare.

**Supervision:** Kefyalew Alemayehu, Solomon Abegaz, Aynalem Haile.

**Validation:** Sisay Asmare.

**Writing – original draft:** Sisay Asmare.

**Writing – review & editing:** Sisay Asmare, Kefyalew Alemayehu, Solomon Abegaz, Aynalem Haile.

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
