## [Decision Letter · Decision Letter 0]

12 May 2021

PONE-D-21-11779

On-farm evaluation of growth and reproductive performances of Washera and Gumuz sheep in northwestern Ethiopia: basics for setting up breeding objectives/goals

PLOS ONE

Dear Dr. Asmare,

Thank you for submitting your manuscript to PLOS ONE. After careful consideration, we feel that it has merit but does not fully meet PLOS ONE’s publication criteria as it currently stands. Therefore, we invite you to submit a revised version of the manuscript that addresses the points raised during the review process.

A careful edition of English must be done. Article present potential however must follow a first round of deep revision.

We look forward to receiving your revised manuscript.

Kind regards,

Carlos E. Ambrósio, Ph.D

Academic Editor

PLOS ONE

Additional Editor Comments:

A careful edition of English must be done. Article present potential however must follow a first round of deep revision.

Journal Requirements:

3. We note that Figure 1 in your submission contain map images which may be copyrighted. All PLOS content is published under the Creative Commons Attribution License (CC BY 4.0), which means that the manuscript, images, and Supporting Information files will be freely available online, and any third party is permitted to access, download, copy, distribute, and use these materials in any way, even commercially, with proper attribution. For these reasons, we cannot publish previously copyrighted maps or satellite images created using proprietary data, such as Google software (Google Maps, Street View, and Earth). For more information, see our copyright guidelines: http://journals.plos.org/plosone/s/licenses-and-copyright.

3.1.    You may seek permission from the original copyright holder of Figure 1 to publish the content specifically under the CC BY 4.0 license. 

3.2.    If you are unable to obtain permission from the original copyright holder to publish these figures under the CC BY 4.0 license or if the copyright holder’s requirements are incompatible with the CC BY 4.0 license, please either i) remove the figure or ii) supply a replacement figure that complies with the CC BY 4.0 license. Please check copyright information on all replacement figures and update the figure caption with source information. If applicable, please specify in the figure caption text when a figure is similar but not identical to the original image and is therefore for illustrative purposes only.

Reviewers' comments:

Reviewer's Responses to Questions

**Comments to the Author**

1. Is the manuscript technically sound, and do the data support the conclusions?

Reviewer #1: Yes

Reviewer #2: Yes

2. Has the statistical analysis been performed appropriately and rigorously? 

Reviewer #1: Yes

Reviewer #2: Yes

3. Have the authors made all data underlying the findings in their manuscript fully available?

Reviewer #1: Yes

Reviewer #2: Yes

4. Is the manuscript presented in an intelligible fashion and written in standard English?

Reviewer #1: Yes

Reviewer #2: Yes

5. Review Comments to the Author

Reviewer #1: Please reviewed manuscript attached file including comments. Moreover, it is better to edit manuscript by a native English person شnd in terms of point, comma, distance, etc., it is better to be carefully corrected and reviewed.

Reviewer #2: General comments:

The manuscript shows the evaluation of performance and reproductive traits in farm conditions for Washera and Gumuz sheep. The results demonstrate the variation between the two breeds evaluated and the sex of the animals, and this interesting information can be used to guide possible breeding programs for sheep in Ethiopia.

The writing format of the text needs revision, adapting to scientific norms.

Specific comments:

Abstract

The division of topics in the abstract is very disproportionate, with the results occupying more than 50% of the text. The conclusion has to be less. The mean values and standard deviation can be excluded and shown only the factors that affected performance and reproductive characteristics. Therefore, it would be important to rewrite lines 28 to 37.

L37-39: Please, delete.

Introduction

The introduction is well written and the main important points for the work have been made. However, it is possible to reduce the text of the introduction by making it more concise.

L48: TLU? Define

L89: on - farm. L91 on farm. L92 on-farm, please use only one.

L97-101: The hypotheses are not well supported by the introduction. I think you can delete this part of the text.

Material and Methods

L163-169: Please, delete

L198-201: Please, delete

L209-210: Please, delete

Results/Discussion

The results and discussion of the present work should be brought together in a single topic. The performance results are presented in table form, so there is no need to describe them in the text. Figure 2 was confused, I believe it can be deleted. Table 1 must not be in the proper format of PlosONE, so the form of visualization must be adjusted and improved.

The presentation of reproductive characteristics should also be in a table format for better visualization, including the probability values, just as it was done with the performance results.

The discussion is good.

L328-340: This part should be in material and methods, please include in the correct topic.

Conclusion

The conclusion is repeating what has already been put in the results.

L447-455: Please, delete

L459-461: Please, delete

6. PLOS authors have the option to publish the peer review history of their article (what does this mean?). If published, this will include your full peer review and any attached files.

Reviewer #1: No

Reviewer #2: No

---

## [Author Response · Author response to Decision Letter 0]

7 Jun 2021

Authors express gratitude to the reviewers for their best comments to improve the quality of the manuscript. We look forward to learn more.

---

## [Decision Letter · Decision Letter 1]

7 Jul 2021

On-farm evaluation of growth and reproductive performances of Washera and Gumuz sheep in northwestern Ethiopia: basics for setting up breeding objectives/goals

PONE-D-21-11779R1

Dear Dr. Asmare,

We’re pleased to inform you that your manuscript has been judged scientifically suitable for publication and will be formally accepted for publication once it meets all outstanding technical requirements.

Kind regards,

Carlos E. Ambrósio, Ph.D

Academic Editor

PLOS ONE

Additional Editor Comments (optional):

Due the quality of the MS and innovative data about this excessive breed and this 3D images, I suggest to publish the MS. I disagree with referee 1 comments because in both round of revision, the MS is proper corrected now.

Reviewers' comments:

Reviewer's Responses to Questions

**Comments to the Author**

1. If the authors have adequately addressed your comments raised in a previous round of review and you feel that this manuscript is now acceptable for publication, you may indicate that here to bypass the “Comments to the Author” section, enter your conflict of interest statement in the “Confidential to Editor” section, and submit your "Accept" recommendation.

Reviewer #1: (No Response)

Reviewer #2: All comments have been addressed

2. Is the manuscript technically sound, and do the data support the conclusions?

Reviewer #1: Yes

Reviewer #2: Yes

3. Has the statistical analysis been performed appropriately and rigorously? 

Reviewer #1: Yes

Reviewer #2: Yes

4. Have the authors made all data underlying the findings in their manuscript fully available?

Reviewer #1: Yes

Reviewer #2: Yes

5. Is the manuscript presented in an intelligible fashion and written in standard English?

Reviewer #1: Yes

Reviewer #2: Yes

6. Review Comments to the Author

Reviewer #1: Unfortunately, the authors did not make some comments and did not answer why they did not.

For example, comment on page 12; Line 359 can be mentioned.

Reviewer #2: (No Response)

7. PLOS authors have the option to publish the peer review history of their article (what does this mean?). If published, this will include your full peer review and any attached files.

Reviewer #1: No

Reviewer #2: No

---

## [Editor Report · Acceptance letter]

12 Jul 2021

PONE-D-21-11779R1 

On-farm evaluation of growth and reproductive performances of Washera and Gumuz sheep in northwestern Ethiopia: basics for setting up breeding objectives/goals 

Dear Dr. Asmare:

I'm pleased to inform you that your manuscript has been deemed suitable for publication in PLOS ONE. Congratulations! Your manuscript is now with our production department. 

Kind regards, 

on behalf of

Dr. Carlos E. Ambrósio 

Academic Editor

PLOS ONE